# Thermodynamic Concept of Water Retention and Physical Quality of the Soil

Andrey V. Smagin [1,2,3]

1   Soil Science Department, Lomonosov Moscow State University, Leninsrye Gory 1-12, 119991 Moscow, Russia; smagin@list.ru
2   Institute of Forest Science of RAS, Moscow Region, Sovetskaya 21, 143030 Uspenskoe, Russia
3   Center for Mathematical Modeling and Design of Sustainable Ecosystems, Peoples' Friendship University of Russia, 117198 Moscow, Russia

**Abstract:** The physical quality of the soil is determined by its interfacial interactions in conditions of variable water content. In this regard, water retention characteristics in the form of functions of water content and its thermodynamic potentials are used as indicators of physical quality and its dynamics in the soil. The combination of centrifugation and thermodesorption methods allowed for the first time the assessment of soil water potentials in the entire range of variation from 0 to $10^6$ J/kg for a representative database (more than 400 samples) of the main genetic types of Eurasian soils, grouped into 5 FAO/USDA soil texture classes. The main fundamental achievement of the research is a physically based diagnosis of the critical values of water content and its thermodynamic potentials that separate the areas of dominance of various forms of soil water, physical forces, and mechanisms of interfacial interactions on the WRC-diagrams of the physical quality of the soil. Theoretical and experimental results of the study are of practical interest of sustainable agronomy for determining the optimal ranges of water content in the soil during plant cultivation, water saving, and salt protection in irrigation, mechanical tillage, and other technological operations.

**Keywords:** soil water content; soil water thermodynamic potentials; surface energy; capillarity and disjoining pressure; WRC-diagrams of soil physical quality; optimal soil water content

## 1. Introduction

In modern agronomy, the physical quality of the soil can be defined as a set of physical properties and processes directly or indirectly controlling the basic environmental and technological functions of the soil in agroecosystems. Most of the properties and processes that determine the physical quality of the soil depend on the interactions of the solid, liquid, and gas phases, as well as on changes in their ratio with variable water content in the soil. The simplest and most common way to assess the physical quality of soil is to study its individual properties in the range of natural variation, followed by grouping into quality indexes with certain environmental standards [1–5]. A more complex thermodynamic concept of the physical quality of the soil is based on the equality of the specific energies (potentials) of the interacting physical phases in the state of thermodynamic equilibrium and uses the water retention curve (WRC) to quantify interfacial interactions in soil with a variable water content (*W*) [4,6–12]. The WRC is also the basis for computer models of energy-mass transfer in soil and, therefore, for predicting the dynamics of its physical quality [13,14].

The gradualist thermodynamic concept and the traditional capillary model of water retention consider the energy state of water in the soil and related physical phenomena (soil consistency, constitution, aggregate stability, deformation by swelling and shrinkage, soil strength, resistance, compression, and compaction during mechanical tillage and load), only as a continuous function of the thermodynamic potential ($\Psi$, [J/kg]) or equivalent soil water pressure (*P*, [kPa]) in the capillary-porous soil system [4,13,15]. An alternative

approach in the form of a classical agronomical concept of soil-hydrological parameters (constants) of water retention assumes the presence of several categories of soil water (adsorbed, tightly bonded and loosely bonded films, capillary, gravitational) with different physical forces and mechanisms of water retention and fixed boundaries of their distribution depending on the water content [4,7,15–20]. Voronin (1984, 1990) first proposed to combine the thermodynamic concept with the doctrine of soil-hydrological constants and developed empirical equations (the "secants" method) for the diagnosis of critical points (soil-hydrological parameters) on the WRC, marking the boundaries between the categories of soil water, different physical forces and mechanisms of water retention, and interfacial interactions (Figure 1). The areas selected by this way on the WRC combine the dominant categories of soil water with different mobility and availability to plants, different functional porosity, rheological state, and resistance to mechanical tillage, which in general allows us to consider such a WRC-diagram as a kind of agrophysical passport of the soil.

Since Voronin's previous works have remained unknown to most European and American specialists, and the only reference to an international publication [7] is not relevant today, it is necessary to explain a number of terms used by him for soil-hydrological parameters of WRC-diagrams (Figure 1). The first area is the dominance of mobile capillary-gravity water, infiltration pores, and the viscous-flow rheological state described by the Voigt and Kelvin model. It is bounded by two characteristic points: the maximum water content in the saturation state ($W_s$) and the capillary water capacity (CW). The matrix potential (pressure) of water in the saturation state is zero, and in the CW state is determined by the empirical equation $\lg|P_{CW}| = 1.17$. The second region is limited by the CW point and the state of the maximum capillary-sorption water capacity (MCSW) with the matrix potential (pressure) depending on the water content, according to the equation $\lg|P_{MCSW}| = 1.17 + W$ (Figure 1). This is the area of predominance of capillary water and aeration pores, as well as volumetric macrocapillary forces that hold water in macropores and inter-aggregate voids. From a physical point of view, the MCSW describes the balance between the macrocapillary forces of water retention and gravity (hydrostatic pressure). In field conditions for homogeneous soils it corresponds to the field water capacity (FW). The next region is a transition from the bulk macrocapillary water retention mechanism to surface forces and from capillary to film water. It is limited by the MCSW point and the maximum molecular water capacity (MMW) or the so-called capillarity rupture point (CRP) in the case of coarse-textured or aggregated three-phased soils, according to [18]. This is an area of film-capillary water, water-conducting pores, and viscoelastic rheological state, according to the Burges and Kelvin model (Figure 1). This small section of WRC between the MCSW and MMW points in agronomical practice characterizes optimal conditions for soil tillage and saving of irrigation water. For the critical point MMW (CRP), the matrix potential (pressure) is determined by the empirical Voronin equation $\lg|P_{MMW}| = 1.17 + 3W$. Beyond this point, when the water content decreases, the soil water is mostly in the form of films (see next region in Figure 1). This rather large area of WRC includes loosely bonded and tightly bonded film water, water-retaining pores, and an elastic-fragile and fragile-elastic rheological state according to the complex Goldstein model. It extends up to the last critical point of maximum adsorption water capacity (MAW), beyond which soil water is only in adsorbed form, and soil exhibits fragile rheological behavior, according to the Hook and Saint-Venant model. This area of relatively low water content is of great importance for assessing the physical quality of the soil and modeling water transport in arid climates [21,22]. The matrix potential (pressure) for the MAW point is calculated using the empirical equation $\lg|P_{MAW}| = 4.2 + 3W$. Additionally, the wilting point (WP) should be included in this list as an important indicator of the physical quality of soils in relation to plants [17,23].

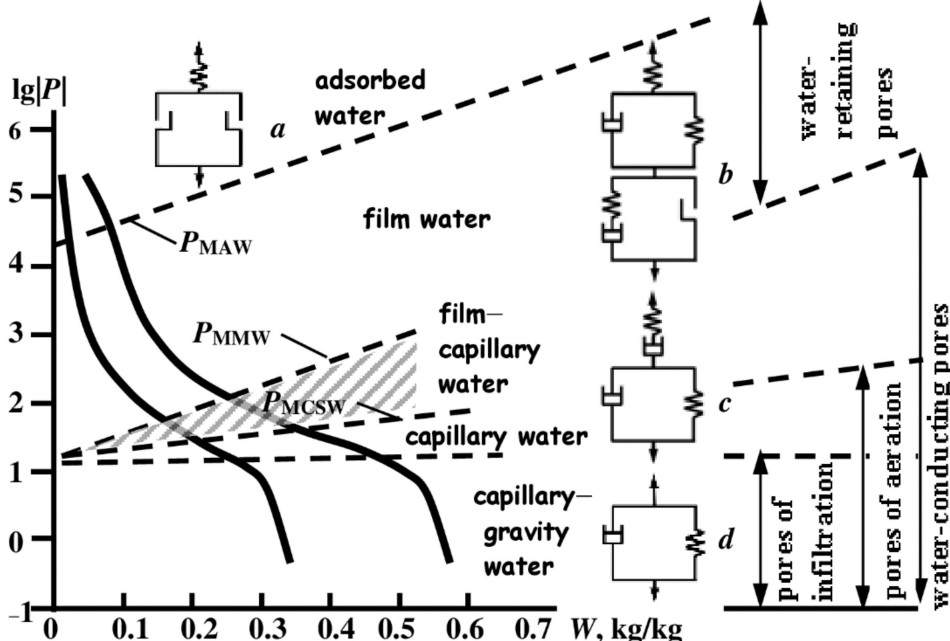

**Figure 1.** WRC as a physical soil passport (according to (Voronin [7]). Dashed lines are the Voronin empirical secants with the following equations: lg | $P_{MAW}$ | = 4.2 + 3 *W*; lg | $P_{MMW}$ | = 1.17 + 3 *W*; lg | $P_{MCSW}$ | = 1.17 + *W*. Rheological models and soil consistency: *a*-fragile state, the Hook and Saint-Venant model; *b*-elastic-fragile and fragile-elastic state, the complex Goldstein model; *c*-viscoelastic state, the Burges and Kelvin model; *d*-viscous-flow state, the Voigt and Kelvin model. The shaded area has optimal conditions for soil tillage and saving of irrigation water.

Developing this approach, Smagin [8,9] gave a complete analysis of the physical forces and mechanisms that control water retention and dispersity dynamics in soils, and also suggested the concept of competitive surface interactions of soil particles with each other and with the liquid phase of a given ionic composition and concentration in the soil physical system, applying to it the fundamental principles and models of the classical DLVO (Deryagin-Landau-Verwey-Overbeek) theory [24,25].

The original thermodynamic direction in soil physics considered together the dynamics of the structure of the pore space and water retention of the soil in the cycles of shrinkage and swelling, in terms of the specific volumes of the physical phases of the soil and its differential porosity [26–31]. Braudeau et al. [30] presented, apparently, the most complete theoretical and methodological development of such an approach. Its practical application in connection with the problem of assessing the physical quality of soils and, in particular, the loss of the structure of chernozems during their compaction was considered in [28,31]. In these studies, along with WRCs, the original diagrams of the structure of the pore space, and quantitative indicators of the surface energy of the solid phase, the critical values of the specific volumes of the pore space with the corresponding standards are used to assess the physical quality of soils with variable pore space.

Dexter [10–12] proposed a universal index of the physical quality of soil in the form of the slope of the pF curve (log*P*(*W*)) at its inflection point according to the van Genuchten WRC empirical model [26], linking it empirically with the most important indicators of soil quality (soil texture, compression, porosity, organic matter content, sodicity, root and water penetration, friability, resistance to mechanical tillage, and surface load). This indicator is calculated by the formula [10]:

$$S_D = n(W_s - W_r)\left(1 + \frac{1}{m}\right)^{-(1+m)} \tag{1}$$

for the inflection point of the pF-curve with coordinates:

$$|P_{IP}| = \frac{1}{\alpha}\left(\frac{1}{m}\right)^{1-m}; \Theta_{IP} = \left(1+\frac{1}{m}\right)^{-m}; W_{IP} = (W_s - W_r)\left(1+\frac{1}{m}\right)^{-m} + W_r \quad (2)$$

where $|P_{IP}|$, $\Theta_{IP}$, $W_{IP}$ are soil water pressure, relative humidity, and water content at the inflection point and $W_s$, $W_r$, $\alpha$, $n$, $m = 1 - 1/n$ are the parameters of the empirical van Genuchten model [32]. The following categories of the Dexter index $S_D$ were suggested: <0.02, very poor; 0.02–0.035, poor; >0.035, good physical quality [12].

Despite its effectiveness, this method, which is also empirical, is problematic from a physical standpoint, since the pF curve is the only way to describe the WRC and their inflection points (of the real curve and the pF curve) do not coincide. The real inflection point for the van Genuchten model is: $|P_{IP}| = \alpha^{-1}(m)^{(1-m)}$; $\Theta_{IP} = (m+1)^{-m}$. In an earlier publication [33] we suggested using this formula to calculate the capillary water capacity of soils and their compositions with swelling hydrogels:

$$CW = (m+1)^{-m})(W_s - W_r) + W_r \quad (3)$$

In general, despite a serious thermodynamic basis, the concept of the physical quality of soil remains largely empirical, and the critical points of water content (soil-hydrological constants) on WRC-diagrams are considered by most experts to be purely conventional values, devoid of exact physical meaning. In this regard, the main purpose of this study was to develop the thermodynamic concept of physical quality of soil with a transition from empirical to physically-based methods for diagnosing critical points of WRC, separating the areas of dominance of various physical forces and mechanisms of interphase interactions that control water retention and physical quality of soils. Their quantitative assessment is based on the traditional capillary model of water retention in the form of the van Genuchten [33] function, as well as on the alternative fundamental exponential model of the disjoining pressure of soil water [9,34], and on the nonlinear dependence of the osmotic pressure (potential) on the soil water content [35]. The new methodological developments, including high-speed centrifugation and thermodesorption of water vapors, made it possible for the first time to obtain and describe WRCs for soils of different genesis and dispersity in the entire range of the absolute values of soil water potential from 0 to $10^6$ J/kg. This range is up to 10,000 times larger than the interval 0–1000 hPa or 0–100 J/kg traditionally used in soil water thermodynamics [26,30,32]. Using this technique, we obtained and analyzed water retention characteristics for a representative database of Eurasian soils (more than 400 samples). The critical soil-hydrological parameters of WRC normalized by the maximum water content ($W_s$) for the main textural classes of Eurasian soils represent the more important practical result of the study. They can be used to determine the boundaries of the optimal water content in different soils in relation to their ecological functions (plant productivity, biodegradation of organic matter, anti-erosion protection of the surface, load-bearing capacity, and protection against salinity) and technological procedures (mechanical tillage, irrigation, and drainage).

## 2. Materials and Methods

In this research, we used our own database of Eurasian soils, published for the first time in [8] and supplemented by subsequent materials [9,29,36]. The WRC database includes more than 400 samples of different texture classes (160 sands, silty sands, and loamy sands; 110 loams, sandy clayey loams, sandy loams, silty loams, clayey loams, and fine loams; 130 silty clayey loams, clayey loams, sandy clays, and clays) of the main genetic types of Eurasian soils according WRB International Soil Classification [37]: Arenosols, Calcisols, Podzols, Luvisols, Retisols, Phaeozems, Umbrisols, Fluvisols, Chernozems, Kastanozems, Solonchaks, Solonetz, Planosols, Vertisols, and Histosols, as well as clay minerals and organogenic porous media, synthetic soil conditioners (hydrogels), and their compositions with mineral soil substrates. The granulometric composition of the soil

was determined by laser diffraction method [38] using a Microtrac S-3000 particle size analyzer (USA).

Thermodynamic analysis of water-retention in soil samples was carried out by combination of equilibrium centrifugation in the author's modification [8] with a new method of soil water thermodesorption [39]. Unlike the well-known Ioffe-Gradner formula used in soil sciences for assessing the matrix pressure (potential) of water during centrifugation, our modification included the gravitational component, which allowed building of the WRC from the state of full saturation:

$$|P_m|, [\text{kPa}] = |\Psi_m|, [\text{J/kg}] = \left\{ 0.011 \cdot n^2 \cdot r \cdot \cos(\alpha) + g \cdot \sin(\alpha) \right\} \cdot h \tag{4}$$

where $n$, rpm is the number of centrifuge revolutions per minute; 0.011 is the conversion factor for the square of the angular centrifuge velocity in $s^{-1}$ calculated from the centrifuge speed in rpm $((2\pi/60)^2 \approx 0.011)$; $r$, [m] is the distance from the axis of rotation to the center of mass of the sample; $h$, [m] is the sample's height; $\alpha$ (in radians) is the angle between the horizontal axis and the central axis of symmetry of the sample; and $|P|$ and $|\Psi|$ are absolute values of soil water pressure and potential in porous media, both having negative signs by definition. The water pressure in the soil in kPa is numerically equal to its thermodynamic potential in J/kg since $|P| = |\Psi| \cdot \rho_\ell$, where the water density $\rho_\ell$ = 1000 kg/m$^3$. We used two high-speed laboratory centrifuges (Hettich Universal 320 (Germany) and CLN-16 (Russia) with water-retention energy ranges (Equation (4)) from 0 to 3030–3689 J/kg (kPa) or to a drier state near maximum hygroscopy at a soil water activity equal to 0.98 ($|\Psi| = -(RT/M)\ln(0.98) = 2734$ J/kg). After the last stage of centrifugation (12,000 rpm), the samples were placed for drying at differential temperature levels from 30 to 105 °C into a KD 200 drying oven (China) with forced circulation and ventilation. This simple procedure estimates the WRC in the range of absolute values of the thermodynamic potential of soil water up to 1,000,000 J/kg, as well as the specific surface area according to the method in [39]. Under the conditions of a thermodynamic state of equilibrium in a laboratory with a constant air humidity ($f$) and temperature ($T_r$), the water potential depends linearly on the temperature in the drying oven ($T$) by the thermodynamic Equation (5), which is obtained from the fundamental Clausius–Clapeyron equation in [39]:

$$\Psi = Q - p \cdot T; \tag{5}$$

where $Q$ = 2401 ± 3 kJ/kg is the specific heat of evaporation for the temperature range of 0–100 °C, $R$ = 8.314 J/(mol·K) is the universal gas constant, $T$ [K] is the absolute temperature in the drying oven, and $M$ = 0.018 kg/mol is the molar mass of water.

At certain stages of centrifugation, soil samples with DS1923 "hygrochron" sensors (USA) implanted were placed in the freezer for 20–30 min. This operation made it possible to determine the total thermodynamic potential of the water in the soil from the temperature ($T_F$) of the "water-ice" phase transition by the formula [9]:

$$\Psi = L \frac{T_F - T_0}{T_0 M} \tag{6}$$

where $L$ = 6013 J/mol is the latent freezing heat for water and $T_0$ = 273 K is the freezing point of pure water. In accordance with the rule of additivity of thermodynamic potentials [7], the difference between the total and matrix soil water potentials represents the osmotic component of soil water pressure ($P_{os}$).

The research also analyzes and summarizes monographic materials published in Russia and the USSR [6,18,30–42] concerning the hydrophysical and technological properties of Eurasian soils in connection with the problem of assessing their physical quality. Computer statistical and mathematical processing, including approximation of experimental data by hydrophysical models and computer simulation of soil water transport, were carried out in the S-Plot 11 program using the «Regression Wizard» toolbox and HYDRUS-1D software [14].

## 3. Results and Discussions

### 3.1. Theoretical Positions

The modern thermodynamic water retention concept [6–9] complements the basic capillary model with representations of surface forces and interactions in the soil physical system in the low to medium ranges of water content, in the form of films and adsorption water layers. This approach allows us to simultaneously study the mechanisms of water-holding capacity (the interaction of solid and liquid phases) and the mutual interactions of the particles of the solid phase, separated by thin layers (films) of water, and hence the mechanical and rheological properties and soil strength. Figure 2 illustrates the balance of film (thin layer) and capillary water (meniscus) or the so-called second kind capillary phenomena [24]. In macro-porous coarse-dispersed objects (Figure 2A) this equilibrium is established directly at the contact of particles due to the interaction of meniscus with negative curvature (R-) and film with positive (R+) and additional to the Laplace ($\pm \Delta P$) disjoining pressure ($P_D$) from the water film to the surface of the solid phase.

More complicated cases occur in polydisperse systems with inert large particles (soil skeleton) and a swelling colloidal-dispersed complex (soil plasma), which are mainly in a two-phase state (gel) in the form of plane-parallel particles separated by thin layers (films) of water (Figure 2B). The overburden lithological pressure and capillary forces (water meniscus) of the soil skeleton restrict the swelling pressure of the colloidal-dispersed complex; the more so, the lower the proportion of colloidal-dispersed particles in the general granulometric composition of the soil. Conversely, if the colloidal-dispersed complex dominates (clayey loams, clays), such soils swell almost unlimitedly, especially their surface layers, passing from a plastic to a viscous-flow state. For such soils, water retentive, rheological, and structural-mechanical properties are controlled mainly by the mechanism of disjoining pressure (Figure 2C) according to Deryagin [24,25] in a wide range of soil water content [8,34,43]. This range corresponds to the stable state of the particles of the colloidal-dispersed complex separated by water films. Theoretically, it is determined by the ratio of the surface molecular adhesion forces (coagulation factor) and the forces of ion-electrostatic and structural repulsion, which protect the particles from coagulation and preserve their high surface energy, which is spent on water retention [8]. Molecular (dispersion) forces of adhesion are inherent in any surface, regardless of their charge. Polar repulsive forces are electrical in nature and arise when the diffuse and adsorption (structured) elements of the electric double layers overlap during the approach of particles with a charged surface [24,25,44]. The fundamental Hamaker-Lifshitz equation [24,44] for the pressure of molecular forces ($P_{mol}$) depending on the thickness of the water film ($h$, m) and the mass content of water ($W$,kg/kg) for the soil physical system with dispersity or specific surface area of particles ($S$, m$^2$/kg) in a first approximation (symmetrical films) can be written as follows [8,21]:

$$P_{mol} = \frac{A_G}{6\pi h^3} = \frac{A_G(S\rho_\ell)^3}{6\pi W^3} \tag{7}$$

where $A_G$, [J] is the generalized Hamaker constant and $\rho_\ell$, [kg/m$^3$] is the density of water.

The fundamental equation of the ion-electrostatic and structural components of the disjoining pressure ($P_D$) in thin films according to Deryagin, modified for the soil physical system, looks like [8,44]:

$$P_D = a \cdot \exp\left(-\frac{h}{\lambda}\right) = a \cdot \exp(-bW), \ b = \frac{1}{S\rho_\ell\lambda} \tag{8}$$

where $\lambda$, [m] is the length of correlation for the structural forces or effective Debye thickness of the double electric layer for ion-electrostatic forces and $a$, [Pa] is the maximum pressure corresponding to the surface charge potential of the particles (h = 0). With the semi-logarithmic coordinates ln($P$) and $W$, Equation (8) transforms into a linear trend. Resurreccion et al. [45] use a similar equation called the Campbell-Shiozawa-Rossi-Nimmo

1992–1994 model, although before these authors it was experimentally discovered in 1948 by Terzaghi in sedimentology and in 1966 by Sudnitsyn in soil science (see the history of this issue in [43]). The physical interpretation of this dependence as a fundamental equation for the ion-electrostatic component of the disjoining pressure by Deryagin in the soil-water physical system was probably first given in [8]. The λ-values near 1 nm (no more than three molecular layers of water) are typical for short-acting structural forces and so-called Newtonian black films or thin α-films [24,44]. Larger λ-values indicate long-range ionic-electrostatic forces that form thick β-films or loosely bounded film water [24,37]. Borchard and Jablonski [46] obtained a thermodynamic exponential dependence of water retention capacity on concentration of a solute in heterophase water systems, which is a similar result if we take into account that concentration is inversely proportional to water content.

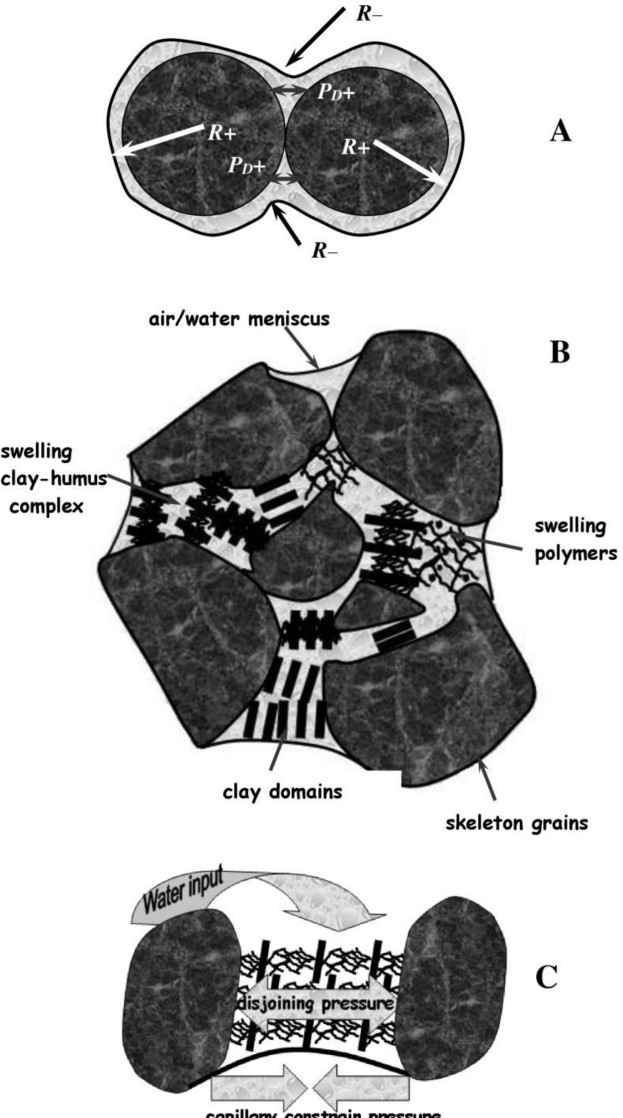

**Figure 2.** The equilibrium pattern of film and capillary water. (**A**) Coarse textured soils, (**B**) swellable soils with a two-phase colloid-dispersed complex (clay domains, swelling polymers, swelling clay-humus complex), (**C**) the effect of disjoining pressure of soil water.

All surface forces controlling interfacial interactions vary with distance from the surface respectively, with the thickness of the water film ($h$) and the water content ($W$) in the porous system with a known dispersity $S$: ($W = h\rho_\ell S$). Therefore, to assess the

physical quality of the soil, it is necessary to consider the balance of physical forces and mechanisms of water retention in the entire range of variation of the water content from 0 to the state of saturation ($W_s$). The point of zero water content ($W = 0$) and the area of strongly bounded water in the vicinity of this point are poorly studied in soil science, despite the importance for arid conditions [21,45]. As a result, for the dominant empirical WRC models, including the van Genuchten model [32], this point is undefined. The potential (pressure) of water in the vicinity of the point $W = 0$ (or $W = Wr$ in the van Genuchten model) tends to infinity, which obviously has no physical meaning. The thermodesorption method proposed in [39] makes it possible to study the energy state of soil water in the vicinity of zero water content. The fundamental thermodynamic Equation (5) in a dry room at a relative humidity $f = 0.2$–$0.3$ and absolute temperatures $T_r = 291$–$293$ K for a standard drying temperature of 105 °C gives the value of the thermodynamic potential of water in the range of 895–999 kJ/kg or near 1 million J/kg. This equation allows us to accurately assess the thermodynamic potential under standard conditions of complete soil dehydration at a temperature of 105 °C instead of the conventional empirical Groenevelt-Grant value (approximately 800 kJ/kg) used in [45].

According to DLVO theory [24,25,44] for the soil physical system, the classical stability criterion for particles separated by an ion-electrostatic and structural barriers can be written through the equality $P_{mol} = P_D$ (7) and (8) and through the equality of the corresponding surface energies $U_{mol} = U_D$, taking into account that $P_{mol} = dU_{mol}/dh$ and $P_D = dU_D/dh$:

$$\left. \begin{aligned} \frac{A_G(S\rho_\ell)^3}{6\pi W^3} &= a \cdot \exp(-bW) \\ \frac{A_G(S\rho_\ell)^2}{12\pi W^2} &= \frac{a}{bS\rho_\ell} \cdot \exp(-bW) \end{aligned} \right\} \tag{9}$$

The solution of Equation (9) gives the following simple condition for the critical soil water content ($W_{cr}$) of the appearance/disappearance of the particle separating energy barrier and mass coagulation of particles of a colloidal-dispersed soil complex:

$$W_{cr} = \frac{2}{b} \tag{10}$$

At $W \leq W_{cr}$, the particles coagulate and the soil physical system acquires minimal dispersity (specific surface area: $S = S_0$). Smagin [9] gives the following expression for estimating $S_0$, based on condition (10) and the minimum thickness of a stable aqueous film in two diameters of water molecules:

$$S_0 = \frac{1}{2br_0\rho_\ell} \tag{11}$$

where $r_0 = 1.38 \cdot 10^{-10}$ m is the crystal-chemical radius of water molecules. The $S_0$ index closely correlates with the BET (Brunauer-Emmet-Teller) estimate of the specific surface according to isotherms of sorption of water vapor [6,9], which also assumes a monomolecular coating of solid particles by water molecules at the corresponding critical humidity point [9,34].

The theoretical formula (11) shows that the WRC slope ($b$) in semi-logarithmic coordinates is determined by the dispersity (effective specific surface) of the soil physical system. This basic position provides, in our opinion, a certain physical meaning to the empirical Dexter index, as the reverse slope of the WRC at the inflection point of the pF curve ($S_D = dW/dlnP$). Recall that many indicators of soil physical quality are closely related to this index [10–12]. Presumably, the Dexter index reflects the state of a maximum interface of physical phases ($Z$, m²/g), which can be determined according to [34]:

$$Z = S_0 b S_D \tag{12}$$

The Dexter index has the dimension $g_w/g_s$, and the slope index WRC $b$ for model (8), respectively, is the inverse dimension $g_s/g_w$; therefore, their product is dimensionless (here

the designations *w, s* refer to the liquid and solid phases of the soil). The study [12] shows that the Dexter index also corresponds to a maximum on the curves of pore size distribution or the state of drainage of pores that dominate in the structure of the soil physical system. Obviously, in this state, the geometric interface must also reach its maximum. Therefore, we can expect that the capillary pressure at the Dexter point will be numerically equal to the maximum height of the capillary rise, and the soil moisture to the value of the field water capacity (FW).

Note that the traditional thermodynamic concept and the capillary model deny the existence of any physically based critical (limited) parameters of water-retention and, in particular, FW [13,20]. In this gradualist concept, the FW value for a homogeneous soil, as the moisture of its surface layer at the boundary with the atmosphere, is a function of the time and height of the soil profile above the level of groundwater standing. This position is illustrated by computer simulation of the FW state during surface waterproofing, and the seepage face of water at the lower boundary of the soil profile (*H*) using HYDRUS-1D software [14] (see Figure 3). The greater *H* (deeper the groundwater), the longer it takes to reach equilibrium and the smaller the FW value. The upshot of this is that the equilibrium distribution of water is the WRC itself, embedded in the software. However, such a common opinion contradicts reality; in particular, the final, strictly fixed for each soil texture class, capillary rise height ($H_{cap}$) with a sharp (by moisture, salts precipitation) boundary of the capillary fringe. For short-profile soils ($H < H_{cap}$), modeling gives quite correct results, emphasizing the dependence of FW not only on the solid-phase matrix (dispersity, structure), but also on the height of the uniform water body that determines hydrostatic pressure [8]. At $H > H_{cap}$, the model ceases to satisfy reality, predicting an essentially infinite, unlimited capillary rise. In nature if $H > H_{cap}$, the WRC-shaped equilibrium imitated by the Richards model (HYDRUS-1D software) is never realized with water rising by tens, hundreds, and thousands of meters (matrix soil water pressures of 100, 1000, 10,000 kPa) with the corresponding transfer of salts. Strictly finite, limited for each soil texture class, the maximum height of capillary rise has a great environmental importance, since it contributes to the conservation of groundwater from evaporation and soil surface from salinization.

The following physically-based formula, in contrast to the well-known Jurin's law, allows a more accurate estimate of $H_{cap}$, [m], taking into account the bulk density of the soil ($\rho_b$) and adsorbed strongly bounded water (MAW) [8]:

$$H_{cap} = \frac{\sigma_\ell S_{sk} \rho_b \cos(\alpha)}{\rho_\ell g(1 - \rho_b/\rho_s - \rho_b \text{MAW}/\rho_\ell)} \tag{13}$$

Here $S_{sk}$ [m$^2$/kg] is the specific surface area of soil particles streamlined by viscous fluid or the so-called soil skeleton surface area; $\rho_s$, [kg/m$^3$] is the density of the solid phase; g, [m/c$^2$] is the acceleration of gravity; $\sigma_\ell$, [N/m] is the surface tension of water at the boundary with air; and $\alpha$ is the wetting angle of the solid phase. The $S_{sc}$ value can be estimated from the differential particle size distribution curves using the formula [8]:

$$S_{sc} = \sum_{0.006}^{1} \left(\frac{6n_i}{\rho_s D_i}\right)/100 = \frac{6}{\rho_s} \int_{0.006}^{1} \frac{f(D)}{D} dD \tag{14}$$

where *D*, is the effective particle diameter; *f(D)* is the differential distribution of granulo-metric elements function obtained experimentally in the range from 0.006 to 1 mm; and $\rho_s$ is the density of the solid phase. Substituting the values of *D* in mm and $\rho_s$ in g/cm$^3$ into formula (14) gives the dimension of $S_{sc}$ in m$^2$/kg, since 1 g/cm$^3$ = 10$^3$ kg/m$^3$. Modern particle size analysis, such as laser diffractometry, determines the function *f(D)* directly. For most of the studied Eurasian soils, from sands to clays and clay loams, the $S_{sc}$ index varied from 5 to 150 m$^2$/kg, or approximately a thousand times lower than $S_0$.

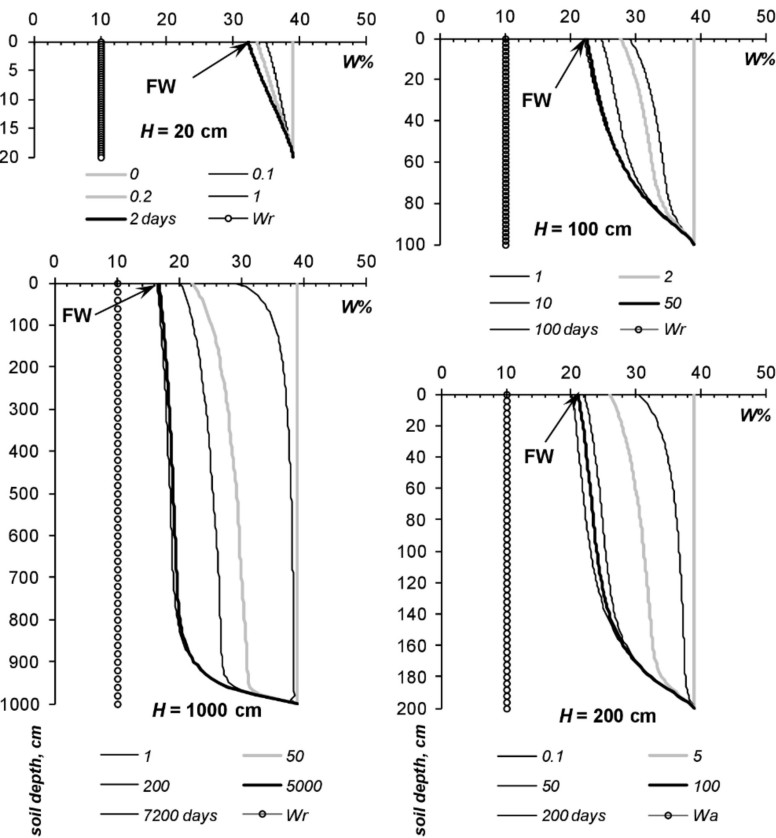

**Figure 3.** Modeling the vertical distribution of water in sandy clayey loam (computer model and WRC database HYDRUS-1D) in equilibrium of capillary forces and gravity (**FW** state). **FW** varies from 33% (**H** = 20 cm) to 16.5% (**H** = 1000 cm), and in deep-profile soil, equilibrium is established for about 20 years (7200 days).

Formula (13) combines the special cases of the well-known classical criteria (according to [15,18]) for determining the critical parameters of water retention in an "ideal" soil of spherical particles with a diameter (*D*), taking into account that $S_{sk} = 6/(\rho_s D)$. For example, taking spherical mineral particles ($\rho_s = 2.6 \cdot 10^3$ kg/m³) with close packing ($\rho_b = 1.5 \cdot 10^3$ kg/m³), MAW = 0, from (13), we obtain the classical equation of capillary rise height, at which film and capillary (meniscus) water coexist: $H_{cap} = 8.2\sigma_\ell/(\rho_\ell g D)$, or appropriate critical pressure (potential) of water: $P_{cr} = 8.2\sigma_\ell/D$, according to [15]. The limiting value of capillary rise must be taken into account in models of the "tipping-bucket" type for a more correct forecast of the water regime and possible unproductive water losses, as well as when calculating moisture-accumulative soil structures, including the green roof [19,47,48]. The maximum thickness (height) of a layer of "pendulous" water above a coarsely dispersed capillary barrier ($H_w$) is the difference between the potential of water in this layer ($\Psi_1$) and in the coarse-textured material of the capillary barrier ($\Psi_2$): $H_w = (\Psi_1 - \Psi_2)/g = H_{cap1} - H_{cap2}$. If $\Psi_2 = 0$ (gravel, crushed stone, expanded clay granules), then $H_w = H_{cap1}$. That is, the thickness of the cultural layer should not exceed the height of the capillary rise of water in its material, otherwise it will lose water by gravitational outflow into the coarse-textured subsoil.

Using Jurin's law ($H_{cap} = 2\sigma_\ell/(\rho_\ell g r)$ it is possible to estimate the critical (limiting) radius of pores (*r*), in which the capillary effect (capillary rise) still exists. For $H_{cap} = 10$ m (theoretical limit based on the maximum allowable water suction at negative pressure of one atmosphere), the limit radius (*r*) is near 1.5 microns. This is the theoretical limit for volumetric (meniscus) macrocapillary forces in soils. The empirical limit is the maximum height of capillary rise in homogeneous loess (near 5–6 m), which gives a pore size of about 6 microns. We used this limiting size in formula (14). At lower sizes, the pores (the

spaces between the solid phase particles) are filled mainly with film water, being under the influence of surface forces with the Deryagin's mechanism of disjoining pressure [24,25].

The stability range of the solvate layers (water films) of the colloidal-dispersed complex is clearly distinguished on the WRC curves as a linear section in semi-logarithmic coordinates, where the positive disjoined pressure mechanism dominates (model (8)). It varies from the first percent of water content in coarse-dispersed soils to 40–60% or more in fine-dispersed soils and clay minerals [9,34]. The lower boundary of the range corresponds to the equality $P_D = P_{mol}$ at the point $W$= MAW (maximum absorption water capacity or non-dissolving water volume in soil, according to [7]. The upper boundary is reached when $P_D = P_{mol}$ is equal at the point $W$ = MMW = CRP (maximum molecular water capacity or capillary rupture point), where water films lose stability under the action of macro-capillary forces and gravity. A physically based experimental determination of the specified boundaries of the stability range is carried out by the function of osmotic potential (pressure) of the liquid phase from its mass fraction in the soil on the basis of a nonlinear physico-statistical model obtained in [35]:

$$F = \frac{P_{os}(W)}{P_{os}^{\max}} = \exp\left[-\left(\frac{\ln\left(\frac{W+k}{MMW+k}\right)}{k \cdot \ln\left(\frac{W}{MAW}\right)}\right)^2\right] \tag{15}$$

Here, $F$ is the dimensionless simulated function of relative osmotic potential ($P_{os}$) represented as the probability density of active concentration of electrolytes in the soil with variable water content ($W$); $P^{max}$ is the maximal values of these functions at the extremum point $W$ = MMW; and $k$ is the empirical parameter controlling the width of the distribution peak. Function (15) is defined mathematically at semi-infinite area MAW< $W$ < ∞, which fits the lognormally distributed $F$-value and the physical essence of soil moisture as any solid/liquid mass ratio on dilution/concentration. In this case, the standardized probability density $F$ changes from zero to one, i.e., remains a finite value not exceeding 1 according to its physical sense. The model (15) explains the presence of a maximum on the thermodynamic curves of $P_{os}$ ($W$) by the interaction of opposite processes of dilution/concentration of free soil solution in the region of capillary and gravitational water and binding of water molecules by the surface of a colloid-dispersed complex in the region of film and adsorbed water. The maximum surface binding deprives the water molecules of the ability to hydrate ions (dissolve substances), which leads to the effect of a non-dissolving volume at $W$ = MAW.

The second part of the study is aimed at experimental verification and confirmation of the above-mentioned theoretical positions in connection with the problem of assessing the physical quality of the soil in sustainable agronomy.

### 3.2. Experimental Results

The diagram of the physical quality of soils shows the universal water-retention curves obtained for I–V groups of the main FAO/USDA texture classes (I-sands, silty sands, II-loamy sands, sandy loams, III-loams, silty loams, sandy clayey loams, IV-silty loams, clayey loams, fine loams, sandy clays, V- clays, silty clays, clayey loams, silty clayey loams) of Eurasian soils of different genesis when processing our database (Figure 4A). The grouping was carried out using scaling by normalizing the water content in the soil with the value $W_s$ (maximum water content in saturation state). The experimental data of the diagram are approximated by the empirical model of van-Genuchten [32] and the fundamental model of disjoining pressure (8), which extends to the linear range of the WRC with the dominance of surface forces of water retention. The evolvent of the scaling diagram using characteristic values of $W_s$ = 23; 32; 44; 61; 62% for five groups of texture classes of mineral soils is shown in Figure 4B. On the vertical axis of diagram 4B, in addition to the absolute values of the matrix pressure (potential) of soil water, the critical pressures are also plotted to estimate soil-hydrological parameters (MSCW or FW, MMW,

WP, MAW) from water retention curves. They are represented by a system of dashed lines in the form of average capillary rise lines ($|P| = H_{cap}$) according to [8], Richards and Weaver's [49] constants ($|P_{FW}| = 33.3$ kPa; $|P_{WP}| = 1500$ kPa), and Voronin's secants ($\lg|P_{MCSW}| = 1.17 + W$; $\lg|P_{MMW}| = 1.17 + 3\,W$; $\lg|P_{MAW}| = 4.2 + 3\,W$), which increase depending on the dispersity of the solid phase [7]. For the scaling diagram (Figure 4A), the Voronin equations give corridors depending on the selected value of saturation water content from $W_s = 20\%$ to $W_s = 60\%$.

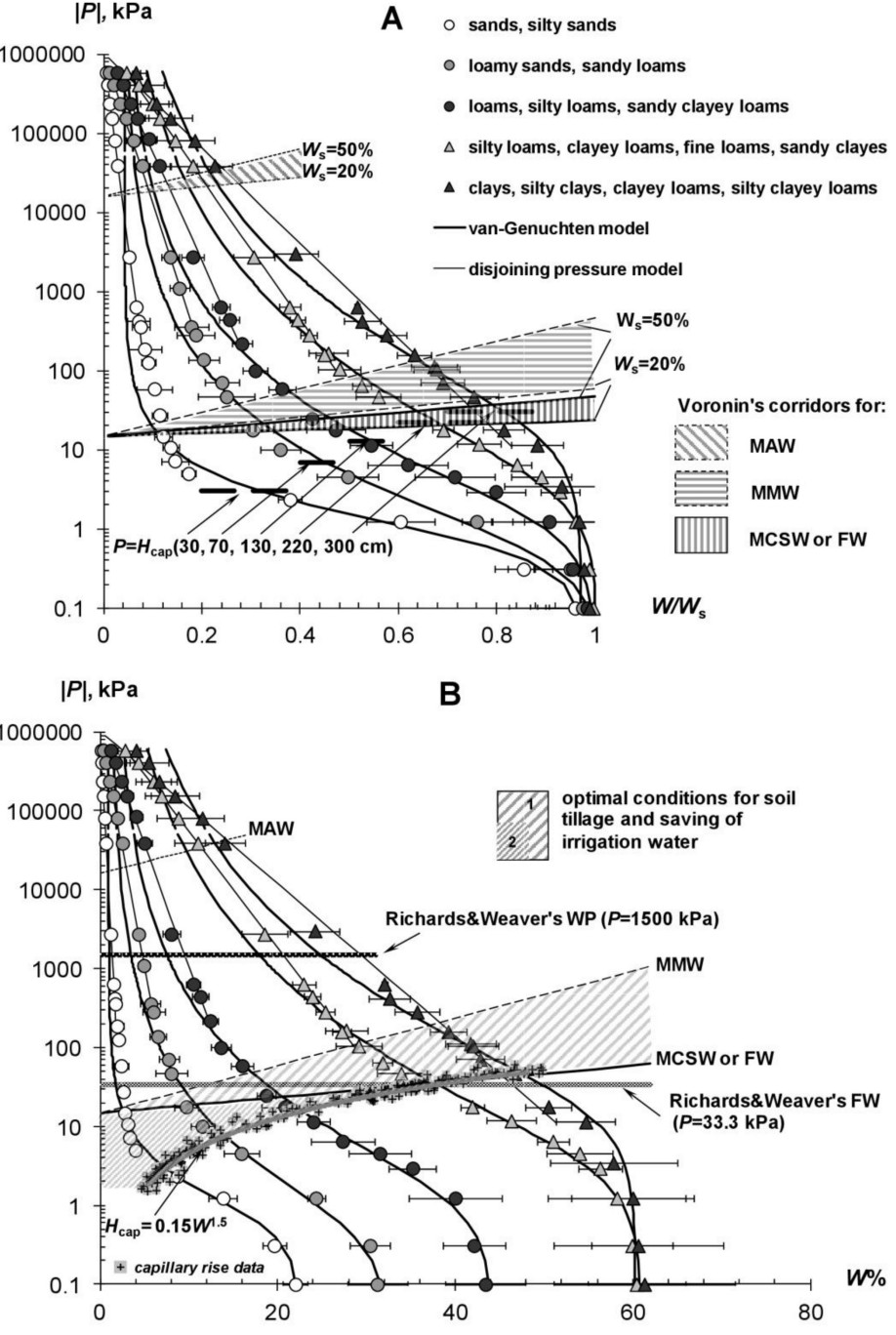

**Figure 4.** Diagrams of the physical quality of Eurasian soils of different genesis and dispersity from the author's WRC database. (**A**) Scaling by using the maximum water content in the saturated state of the soil ($W/W_s$), (**B**) evolvent of the scaling diagram using typical values of $W_s = 23, 32, 44, 61, 62\%$; horizontal bars are confidence intervals at $p < 0.05$.

An analysis of the physical quality diagrams of soils Figure 4A,B reveals a regular shift of WRCs in the direction of increasing the soil water energy (potential) and soil water capacity during the transition from the coarse-textured classes of groups I, II (sands, loamy sands) to finely dispersed soils of groups IV and V (loams, clays). This result is clearly visible from a comparison of WRCs, as well as from a comparative analysis of the slope of the WRCs (*b*) and their inverse values of soil dispersity, estimated by formula (11) (Table 1). The specific surface area ($S_0$) varies from $7.7 \pm 1.4$ in the first group of sands to $162.9 \pm 20.5$ m$^2$/g in the fifth group, which combines clayey loams and clays. The Dexter's index (1) of soil physical quality increased from 0.056 in group II to 0.076 in group IV–V with average values of 0.067–0.073 in groups IV and I. In all cases, it was above the boundary of 0.035; that is, it indicated a good physical quality of all studied Eurasian soils. This result most likely indicates the dependence of Dexter's index on instrumental methods of obtaining the WRC, since it is difficult to assume that all 400 studied samples of Eurasian soils were better than their European counterparts analyzed in [10–12]. The index ($Z$) of the maximum phase interface (12), varied from 204 m$^2$/g (group II) to 277 m$^2$/g (group IV) with an average value of $252 \pm 30$ m$^2$/g (Table 1). It allowed us to estimate the fraction of the interface between the liquid phase and air ($S_{w/a} = Z - S_0$) at the critical point with maximum dispersity of the soil physical system, which, apparently, can be regarded as the extensive contribution (by specific area) of the capillary mechanism to water retention. Its relative contribution ($S_{w/a}/Z$) gradually decreased with increasing soil specific surface area and ranged from 97% in sands (group I) to 39% in clays and clayey loams (group V).

**Table 1.** Estimation of parameters in water retention and dispersity models for five groups of main textural classes of Eurasian soils.

| Soil Texture Classes; Parameters | I Sa, SiSa | II LSa, SaL | III L, SaClL, SaL | IV SiL, ClL, FL | V Cl, SiCl, SaCl, ClL, SiClL |
|---|---|---|---|---|---|
| | The van Genuchten [32] model | | | | |
| $W_s\%$ | $22.3 \pm 0.5$ | $32.4 \pm 0.7$ | $44.1 \pm 0.7$ | $60.9 \pm 0.8$ | $60.3 \pm 0.9$ |
| $W_r\%$ | $1.0 \pm 0.6$ ($p = 0.03$) | $1.3 \pm 0.5$ ($p = 0.014$) | $1.6 \pm 0.7$ ($p = 0.058$) | $0.0 \pm 1.7$ ($p = 1$) | $0.0 \pm 3.7$ ($p = 1$) |
| $\alpha$, kPa$^{-1}$ | $1.33 \pm 0.21$ | $1.81 \pm 0.44$ | $0.71 \pm 0.14$ | $0.29 \pm 0.06$ | $0.06 \pm 0.02$ ($p = 0.004$) |
| $n$ | $1.85 \pm 0.10$ | $1.33 \pm 0.03$ | $1.28 \pm 0.02$ | $1.20 \pm 0.02$ | $1.20 \pm 0.04$ |
| $S_D$, g/g | $0.073 \pm 0.010$ | $0.056 \pm 0.006$ | $0.067 \pm 0.006$ | $0.076 \pm 0.005$ | $0.075 \pm 0.010$ |
| $Z$, m$^2$/g | $265.9 \pm 90.1$ | $203.9 \pm 49.0$ | $242.1 \pm 70.9$ | $277.5 \pm 70.1$ | $268.8 \pm 119.8$ |
| $S_{w/a}/Z,\%$ | $97.1 \pm 0.7$ | $87.2 \pm 2.4$ | $79.0 \pm 3.4$ | $61.3 \pm 9.1$ | $39.4 \pm 12.9$ |
| | Fundamental model of disjoining pressure (Equation (8)) by Smagin [8,34] | | | | |
| $a$, J/kg | $9.3 \cdot 10^5 \pm 1.1 \cdot 10^5$ | $1.1 \cdot 10^6 \pm 8.8 \cdot 10^4$ | $1.3 \cdot 10^6 \pm 2.0 \cdot 10^5$ | $1.6 \cdot 10^6 \pm 1.4 \cdot 10^5$ | $1.0 \cdot 10^6 \pm 2.0 \cdot 10^5$ |
| $b$,% $^{-1}$ | $4.73 \pm 0.09$ | $1.39 \pm 0.02$ | $0.71 \pm 0.01$ | $0.34 \pm 0.03$ | $0.22 \pm 0.04$ |
| $S_0$, m$^2$/g | $7.7 \pm 1.4$ | $26.2 \pm 3.1$ | $50.9 \pm 9.6$ | $107.4 \pm 10.5$ | $162.9 \pm 20.5$ |

Sa = sand(y), L = loam(y), Si = silty, Cl = clay(ey), F = fine; *p*-is the level of significance; if not indicated in parentheses, then $p < 0.001$; $\pm$ indicates the corresponding confidence intervals.

The intensive contribution of capillarity (by energy of water retention) was found to be much lower in comparison with the surface mechanisms. A consistent increase in particle dispersity ($S_0$) and associated surface energy leads to an expansion in the range of dominance of surface water retention mechanisms and, accordingly, to an increase in the linear section of the WRC that satisfies the disjoining pressure model (8) (Figure 4A). If in sand the linear range is limited by water pressures $|P| > 1000$ kPa and water content no more than $0.1\ W_s$, then in fine-dispersed texture groups IV and V it expands to absolute values of pressure 10–100 kPa and water content 0.6–0.85 Ws, i.e., the main amount of water in the soil. This important result, as well as the analysis of the phase interface, indicates the limited range of the traditional capillary model of water retention in assessing the physical quality of soils and the need to supplement it with other mechanisms (models) that take into account the action of surface forces [8,21,43].

The soil-hydrological parameters (critical points) marking the boundaries between the categories of soil water, different forces, and mechanisms of water retention and interfacial interactions are given in Table 2. Their sequential analysis for the studied Eurasian soils gives the following results. The MCSW state or the corresponding FW value varies from 0.1–0.3 $W_s$ in the first group to 0.7–0.8 $W_s$ in the fifth group of texture classes of Eurasian soils. Its assessment by Richards and Weaver's [49] and Voronin's [7] methods gives underestimated results for the first two groups (I, II) of coarse-textured soil classes (sands, sandy loam, loamy sands) compared to the real values of field capacity observed in field experiments (FW real). The estimation method [8] using the average capillary rise height ($H_{cap}$) gives more accurate results for these soil texture classes that are close to real field data. Using the estimate of $H_{cap}$ from the data on the particle size distribution (Equations (13) and (14)), we obtained for a representative sampling ($n = 243$) an empirical equation connecting critical water pressure $P_{cr} = H_{cap}\rho_\ell$ and the water content at the point of intersection with the WRC:

$$H_{cap}, [\text{dm}] = P_{cr}[\text{kPa}] = 1.5 \cdot W^{3/2} \tag{16}$$

where $W$, [%] (per mass). From our point of view, this formula makes it possible to determine the value of the field water capacity from water retention curves better than the Voronin equation ($\lg |P_{cr}| = 1.17 + W$) or Richards and Weaver's line ($P_{cr} = 33.3$ kPa), especially for coarse-textured soils (groups I, II), where the approaches of Voronin [7] and Richards and Weaver [49] give a noticeable underestimation of FW. Experimental data of $H_{cap}$ assessment in [dm] are shown in Figure 4B with cross symbols, which are connected by line (Equation (16)). The figure clearly shows the discussed discrepancies in the estimates of $P_{cr}$ and FW for coarse-textured soils. In fact, a suction of 15–33 kPa ($P_{cr}$ estimate by Voronin or Richards and Weaver) removes practically all liquid water in the sands, leaving no more than 1–2% of the water content. Our approach, based on the height of capillary rise, or the real balance of macrocapillary forces that retain water and the force of gravity that removes water from the soil, gives quite adequate reality FW-values for sands of about 5–6% (Figure 4B). Table 2 also contains averaged data on critical values of inflection point pressure of pF curves ($P_{IP}$) for 12 FAO/USDA soil texture classes from Dexter [12]. The $P_{IP}$ values, apparently, are in full agreement with the average statistical heights of the capillary rise of water, which, in our opinion, confirms the hypothesis, about the maximum manifestation of capillarity in the MCSW state, expressed in the «Theoretical» section.

The cessation of macroscopic mass transfer of water and dissolved substances in the soil is limited by the critical moisture of the soil, which in the Russian hydrophysical school [6,18] was called the capillarity rupture point (CRP) or the maximum molecular water capacity (MMW) identical to it. The physical significance of the capillarity rupture effect consists in the disappearance of capillary phenomena with the dominance of macro-capillary forces with negative meniscus curvature and the emergence of a stable meniscus/film equilibrium (capillary phenomena of the second kind, according [24]), sharply limiting the mobility of soil water, along with the strong binding of water in the films of two-phase colloid-dispersed complex, where capillarity is absent at all (see Figure 2). Voronin [7] suggested the empirical equation $\lg |P_{MMW}| = 1.17 + 3W$ to determine the point of capillarity rupture or MMW by WRC. The physically based approach in [29] uses for this purpose a physical-statistical model (15) for an experimental osmotic pressure curve as a function of the water content in the soil $P_{os}(W)$. The CRP (MMW) is the value of soil water content at the extremum point (maximum) on this curve. Figure 5 illustrates this approach by the example of soils of different genesis and dispersity, from sandy and sandy loam mineral soils to organic colloidal-dispersed system in the form of peat. In many cases, the maximum on the $P_{os}(W)$ curves is close to the Voronin crossover point (Figure 5). It is important to note that the MMW value is determined not only by the solid, but also by the liquid phase of the soil. Thus, MMW in the colloidal-dispersed system with the dominance of the ion-electrostatic mechanism of water retention (Figure 5E,F) decreases with an increase in the concentration of the solution ($C$) due to a decrease in the Debay width of the double electric layer ($\lambda \sim (C)^{-0.5}$) and a partial loss of stability of fine

particles separated by water films. This circumstance should be taken into account in the well-known alternative method [7,18] for assessing CR (MMW) by stopping the transfer of saline solutions to the evaporating surface of the soil sample (MMWreal in Table 2).

**Table 2.** Scaling indicators of physical quality for the main textural classes of Eurasian soils.

| Soil Texture Classes; Soil Quality Indicators | I Sa, SiSa | II LSa, SaL | III L, SaClL, SaL | IV SiL, ClL, FL | V Cl, SiCl, SaCl, ClL, SiClL |
|---|---|---|---|---|---|
| $W_s\%$ | $23.0 \pm 7.6$ | $32.1 \pm 3.8$ | $44.1 \pm 17.1$ | $53.5 \pm 13.8$ | $55.1 \pm 15.1$ |
| **FW or MCSW (dimensionless)** | | | | | |
| FW/$W_s$ by Richard-Weaver [49] ($|P| = 33.3$ kPa) | $0.11 \pm 0.02$ | $0.30 \pm 0.04$ | $0.46 \pm 0.04$ | $0.67 \pm 0.06$ | $0.77 \pm 0.06$ |
| FW/$W_s$ by Voronin [7] (lg$|P| = 1.17 + (W/W_s)W_s$ | $0.13 \pm 0.02$ | $0.34 \pm 0.04$ | $0.47 \pm 0.04$ | $0.66 \pm 0.04$ | $0.80 \pm 0.06$ |
| $H_{cap}$, cm | $30.0 \pm 20$ | $70.0 \pm 28$ | $130 \pm 35$ | $220 \pm 64$ | $300 \pm 35$ |
| FW/$W_s$ by Smagin [8] ($|P| = H_{cap}$) | $0.30 \pm 0.03$ | $0.43 \pm 0.03$ | $0.52 \pm 0.03$ | $0.69 \pm 0.02$ | $0.80 \pm 0.02$ |
| $|P_{IP}|$, in cm by Dexter [10] | $37 \pm 12$ | $64 \pm 26$ | $141 \pm 117$ | $267 \pm 200$ | $272 \pm 130$ |
| FW/$W_s$ by IP of WRC ($|P| = |P_{IP}|$) by Dexter [10] | $0.58 \pm 0.04$ | $0.55 \pm 0.09$ | $0.54 \pm 0.03$ | $0.62 \pm 0.04$ | $0.68 \pm 0.13$ |
| FW/$W_s$ real | $0.28 \pm 0.09$ | $0.45 \pm 0.08$ | $0.54 \pm 0.06$ | $0.69 \pm 0.05$ | $0.81 \pm 0.06$ |
| **MMW or capillary rupture point (dimensionless)** | | | | | |
| MMW/$W_s$ by Voronin [7] (lg$|P| = 1.17 + 3(W/W_s)W_s$ | $0.13 \pm 0.02$ | $0.28 \pm 0.04$ | $0.40 \pm 0.06$ | $0.53 \pm 0.06$ | $0.64 \pm 0.06$ |
| MMW/$W_s$ by model (15) | $0.10 \pm 0.02$ | $0.27 \pm 0.02$ | $0.34 \pm 0.05$ | $0.52 \pm 0.04$ | - |
| MMW/$W_s$ real | $0.12 \pm 0.09$ | $0.24 \pm 0.05$ | $0.32 \pm 0.03$ | $0.47 \pm 0.03$ | $0.59 \pm 0.04$ |
| $W_{opt}$/$W_s$ real | – | $0.28 \pm 0.02$ | $0.37 \pm 0.03$ | $0.48 \pm 0.03$ | $0.61 \pm 0.02$ |
| **WP (dimensionless)** | | | | | |
| WP/$W_s$ by Richard-Weaver [23] ($|P| = 1500$ kPa) | $0.06 \pm 0.02$ | $0.14 \pm 0.03$ | $0.22 \pm 0.03$ | $0.32 \pm 0.02$ | $0.43 \pm 0.04$ |
| WP/$W_s$ by Shaw [50] (WP = $0.85 + 0.96$ WP$_{1500}$) | $0.09 \pm 0.02$ | $0.16 \pm 0.03$ | $0.23 \pm 0.03$ | $0.32 \pm 0.02$ | $0.43 \pm 0.04$ |
| WP/$W_s$ real | $0.09 \pm 0.04$ | $0.18 \pm 0.04$ | $0.24 \pm 0.04$ | $0.33 \pm 0.04$ | $0.49 \pm 0.06$ |
| **MAW or non-dissolving water volume (dimensionless)** | | | | | |
| MAW/$W_s$ by Voronin [7] (lg$|P| = 4.2 + 3(W/W_s)W_s$ | $0.04 \pm 0.01$ | $0.09 \pm 0.02$ | $0.12 \pm 0.02$ | $0.18 \pm 0.04$ | $0.24 \pm 0.04$ |
| MAW/$W_s$ by model (15) | $0.02 \pm 0.01$ | $0.11 \pm 0.04$ | $0.13 \pm 0.04$ | $0.21 \pm 0.08$ | - |

Sa = sand(y), L = loam(y), Si = silty, Cl = clay(ey), F = fine; $\pm$ indicates the corresponding confidence intervals at a significance level $p \leq 0.01$; «real» indicates summary data from monographic issues in USSR and Russia [6,18,40–42].

Voronin [7] apparently for the first time connected the state of MMW with the optimum tillage moisture ($W_{opt}$), technologically important for tillage, with minimal soil resistance and the formation of agronomically valuable granular aggregates. From his point of view, in this state, the free energy of the soil physical system is determined mainly by the surface area of water films, which in mechanical tillage tends, together with a portion of particles, to take the form with the smallest surface, that is, spherical. Additionally, the disappearance of capillary continuity and meniscus contacts will help to reduce the resistance of the soil to mechanical treatment. We have generalized classical materials [40] investigating soil resistivity for tillage in the main types of Eurasian soils of different texture classes (Figure 6A–D). The resulting scaling values ($W_{opt}/W_s$), as Table 2 shows, do not differ statistically from MMW, which confirms the assumption of Voronin [7] about their identity.

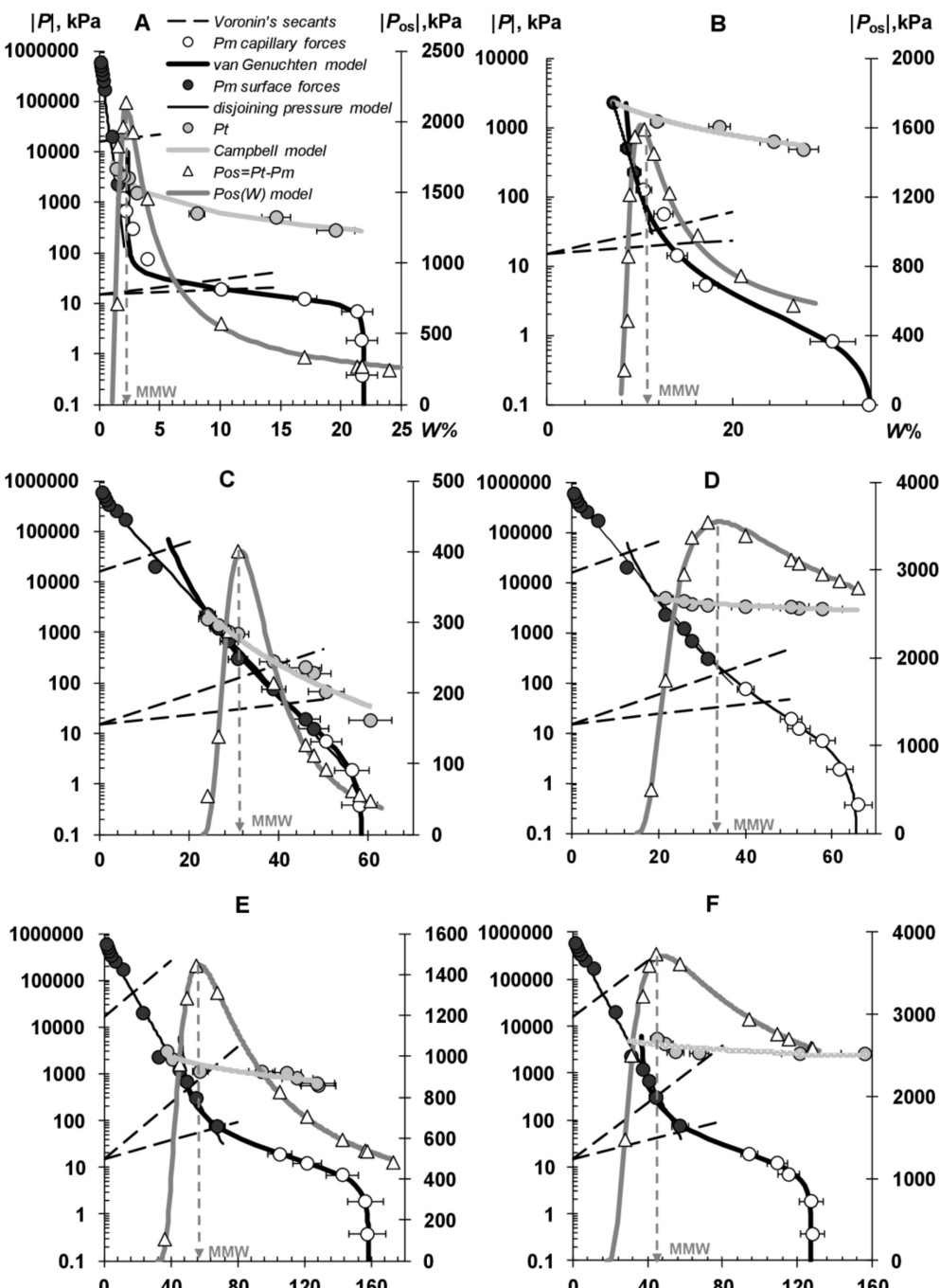

**Figure 5.** Comparison of total $P_t$, matrix $P_m$, and osmotic $P_{os}$ soil water potentials (pressures) in some Eurasian soils of different genesis. (**A**) Sandy Arenosol (Dubai), (**B**) loamy sandy Calcisols (Astrakhan region), (**C**,**D**) clayey loamy Chernozem (Lipetsk region), (**E**,**F**) Eutric Histosol (Moskow region); (**C**,**E**) saturation in distilled water, (**D**,**F**) saturation in 0.5 M NaCl.

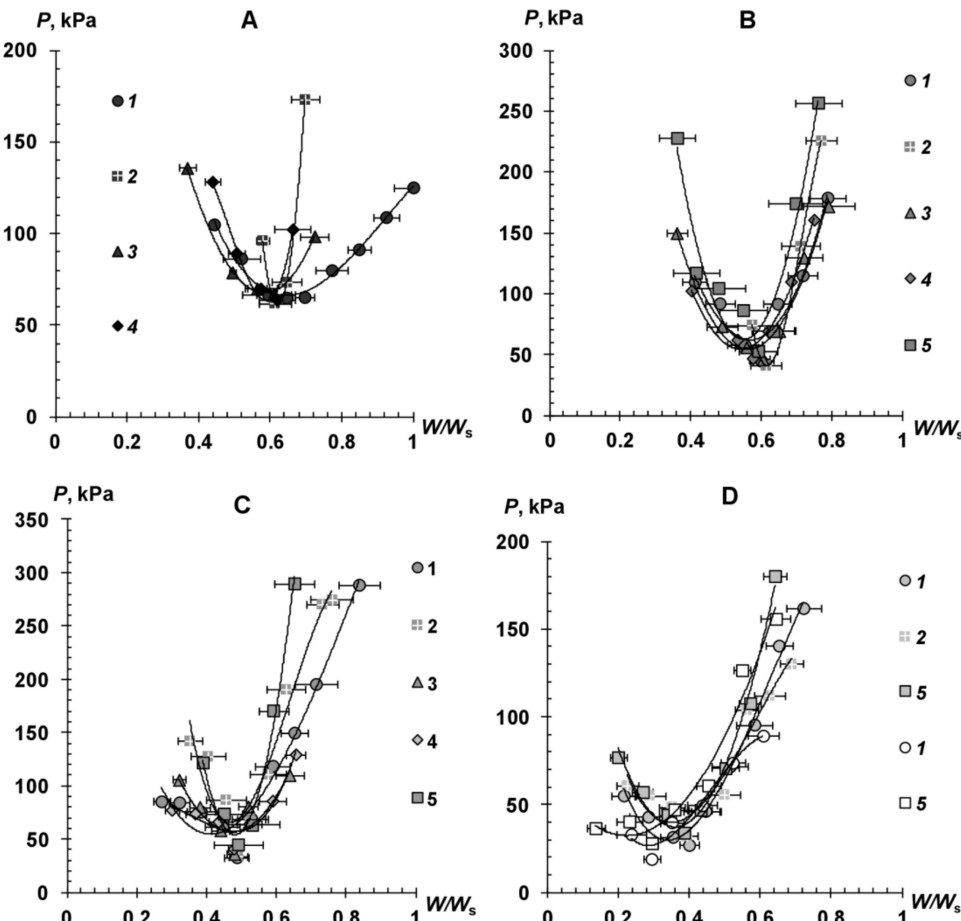

**Figure 6.** Soil resistivity to tillage (*P*, kPa) depending on relative water content ($W/W_s$) for different texture classes of Eurasian arable soils (author's processing of experimental data [40]). Soil texture classes: (**A**) Cl, (**B**) SiL, ClL, (**C**) L, SaClL, (**D**) LSa, SaL (the intensity of symbols color falls from clays (black) to loamy sands (colorless)); Soil types: *1*-Podzols, Luvisols, *2*-Chernozems, *3*-Kastanozems, *4*-Alisols, *5*-Phaeozems, Umbrisols.

Figure 7A shows a close correlation between the Voronin predicted method ($MMW_F$) and experimental values of the maximum molecular water capacity ($MMW_R$) determined by $P_{os}(W)$ curves, according to [35]. The best fit arises when the WRC disjoining pressure model (8) is used for Voronin's calculation, while the van-Genuchten model often gives an overestimated MMW. The FW-MMW range highlighted by shading on the soil physical quality diagrams (Figures 1 and 4B) corresponds to optimal conditions for soil tillage, as well as for saving water resources in irrigation agriculture, since water in this range is still available to plants and has a maximum concentration of nutrients (maximum osmotic pressure) with minimal unproductive losses to subsoil runoff and evaporation. We extended this range with respect to the Voronin approach (Figure 1) using the FW estimate by the capillary rise height (Figure 4B). This expansion affected to a large extent I-III groups of soil texture classes. It is shown by the shaded area (1) against the background of the Voronin area (2) in Figure 4B.

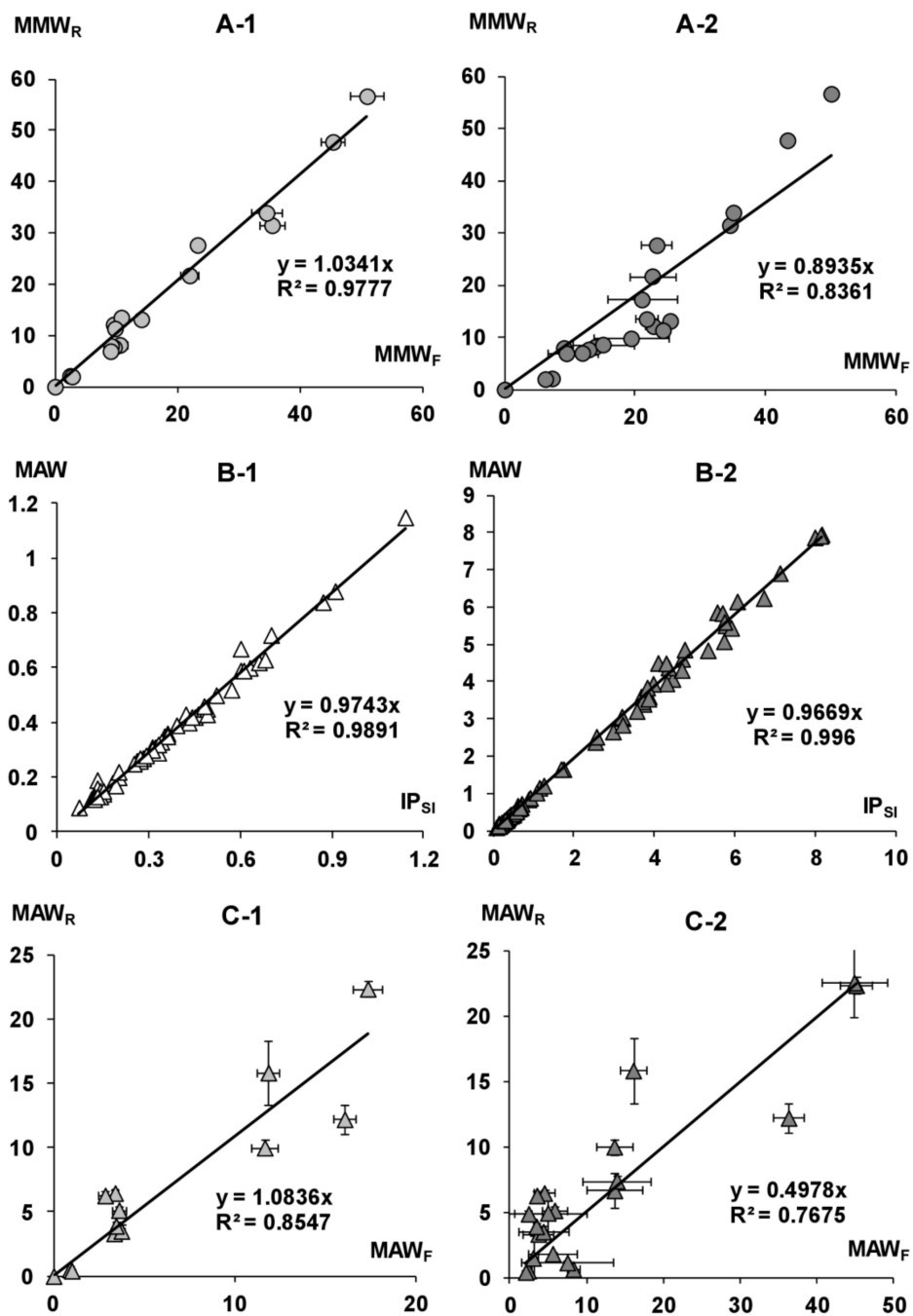

**Figure 7. A**: Correlation of Voronin's predicted and real MMW values: (**A-1**)-fundamental model of disjoining pressure (8), (**A-2**) the van Genuchten model. **B**: Correlation of Voronin's predicted MAW value and water vapor sorption isotherms inflection points ($IP_{SI}$): (**B-1**) coarse-textured soils (58 samples), (**B-2**) all soil texture classes (105 samples). (**C**): Correlation of Voronin's predicted and real MAW values: (**C-1**) disjoining pressure model, (**C-2**) the van Genuchten model.

The next technologically important soil-hydrological parameter is the wilting point (WP). Obtaining real data ($WP_{real}$ in Table 2) requires tedious laboratory experiments (method of vegetative miniatures) or lengthy field observations of plant growth in drought conditions. Therefore, indirect methods for assessing WP by WRC are relevant for studying the physical quality of soils in relation to plants. Traditionally, WP is determined by the Richards-Weaver [23] method at an absolute value of soil water pressure $|P| = 1500$ kPa (Figure 4B). As Table 2 shows, this calculation method gives an adequate estimate for finely

dispersed soils (group III-V of soil texture classes), and in coarse-texture soils (group I-II) it systematically underestimates WP values. The Richards-Weaver method can be improved by using the WP correction for the empirical equation [50]: WP = 0.857 + 0.96 WP$_{1500}$. In this case, the predicted values coincide with the actual data for all five groups of texture classes of Eurasian soils.

The last critical point on the diagram of soil physical quality is the maximum adsorption water capacity (MAW) or a close to it indicator of the non-dissolving volume of water in the soil. They have important theoretical significance as the boundaries of strongly holding stationary water, not capable of dissociating salts in the soil. Their practical value consists in their use in modeling water and solute transport, shrinkage and swelling phenomena, stability and strength of aggregates, in estimating dispersity, sorption and ion exchange in soils, as well as in some pedotransfer functions for WRC. Voronin [7] proposed the empirical equation lg|$P_{MAW}$| = 4.2 + 3 $W$ to estimate the value of MAW by WRC. We assumed [8] that physically MAW is an inflection point on the water vapor sorption isotherm or the boundary dividing the regions of classical Langmuir adsorption and polymolecular water vapor sorption along with microcapillary condensation, where the curve of the sorption isotherm rises upward with small changes in water vapor pressure. A correlative analysis of the relationship between the MMW estimate by the Voronin method and the inflection point on the sorption isotherms from the author's database of Eurasian soils confirms the complete identity of these values (Figure 7B).

Table 2 indicates the statistically significant (within confidence intervals) convergence of the MAW estimates by the Voronin method and experimental estimates by $P_{os}(W)$ curves using model (15) for groups II, III, and IV of soil texture classes. An exception is group I (sands, silty sands), where the estimation according to model (15) gives a half-value of MAW, which may be due to the problem of the accuracy of determining the total soil water potential by the cryoscopic method in coarse-textured soils with a very low water content. Similarly, to the MMW value, MAW must depend not only on the solid but also on the liquid phase, whose composition and concentration affect the Debay width of the double electric layer [24,25,44]. Figure 5E,F confirms this theoretical assumption. Here, the MAW value 12.2 ± 0.1% for a colloidal dispersed peat sample with a salted liquid phase is almost half that in a sample saturated with distilled water (22 ± 0.6%). Comparison of MAW estimation by the Voronin [7] method using different models shows the advantage of estimating the MMW according to the fundamental model (8), when compared to the van-Genuchten model [32]; however, the correlation between calculated and real data was poorer (Figure 7C). The van Genuchten model and its analogs with power functions of pressure and water content in soil are not very suitable for describing water retention in the region of film and adsorbed water. Therefore, the parameter $Wr$, with which a number of researchers associate strongly bounded water, is statistically insignificant for most soils (0.06 ≤ $p$ ≤ 1, see Table 1). An alternative exponential model (8) is more suitable for describing water retention and determining the parameters of the physical quality of soils in this area, which is also confirmed by other researchers [43,45].

The water content directly determines the relationship between the forces and mechanisms of water retention and other interfacial interactions that control the physical quality of soils. In contrast to the pressure (potential) of water, the water content in the soil can be easily determined over the entire range of variation (0 ≤ $W$ ≤ $W_s$); moreover, modern instrumental methods allow monitoring soil moisture automatically. Therefore, in practice it is convenient to use the relative soil water content index ($W/W_s$), normalized by the maximum value in the state of saturation, in order to characterize the physical quality of soils. The $W_s$ values were determined by the well-known formula [6,13]:

$$W_s = \frac{\rho_\ell}{\rho_b} - \frac{\rho_\ell}{\rho_s} \tag{17}$$

using experimental information on soil density ($\rho_b$) and the density of its solid phase ($\rho_s$). Complementing our previous work [51], with the results obtained in this study, it is possible

to propose the following table for assessing physical quality according to the $W/W_s$ index in connection with the main ecological functions and services of soils (plant productivity, biodegradation and preservation of the soil metagenome, absorption of precipitation and underground water recharge, protection of the surface from water and wind erosion, protection from salinization and landslides, improvement of load-bearing capacity for buildings and equipment, reduction of resistance to mechanical processing, etc.) (Table 3). The estimation of the upper limit of the water content at which anaerobiosis and plant depression occur due to lack of air was based on the value of the air input pressure $(1/\alpha)$ according to the van Genuchten model and on the CW indicator (see Formula (3)). Other boundary values of the $W/W_s$ index were obtained, taking into account the results of this study (Table 2). The standards of the physical quality of soils proposed on the basis of a thermodynamic approach generalize a representative database on water retention for the main textural classes of Eurasian soils of different genesis. Unlike the well-known empirical approaches [1–5,10–12], our development covers the entire range of water content in the soil and is based on a fundamental analysis of the physical forces and mechanisms controlling the physical quality of the soil. This development, we hope, will be useful in the practice of ecological assessment of the physical quality of soil and optimization of its ecological functions and services, especially in arid irrigated agriculture, where maintaining the optimal range of water content (grey hatch in Table 3) will effectively save water resources and protect the soil from secondary salinization [4,7,8].

**Table 3.** Soil physical quality standards using $W/W_s$ index.

| Five Groups of Main FAO/USDA Soil Texture Classes | | | | | I-Sands, Silty Sands, II-Loamy Sands, Sandy Loams, III-Loams, Silty Loams, Sandy Clayey Loams, IV-Silty Loams, Clayey Loams, Fine Loams, Sandy Clays, V-Clays, Silty Clays, Clayey Loams, Silty Clayey Loams. |
|---|---|---|---|---|---|
| I | II | III | IV | V | |
| Physical Quality Index $W/W_s$ | | | | | Environmental Comments: |
| >0.90 | >0.90 | >0.85 | >0.85 | >0.85 | High non-productive losses (infiltration, evaporation) grow depression by the lack of air in the soil (over-wetting) up to the death of plants during prolonged over-wetting, anaerobiosis and root rot, suppression of basal respiration and rapid (aerobic) biodegradation of organic matter, emission of toxic greenhouse gases (methane, nitrous oxide, hydrogen, hydrogen sulfide etc.), viscous-flow state with loss of load-bearing capacity and linking of wheeled vehicles, high risks of landslides and water erosion. |
| 0.25–0.9 | 0.45–0.90 | 0.55–0.85 | 0.70–0.85 | 0.80–0.85 | Optimal for high plant productivity easily accessible soil water, but high nonproductive losses (infiltration, evaporation), maximum biological activity and intensity of organic biodegradation, risk of excessive release of carbon dioxide greenhouse gas, high hydraulic conductivity and capillary rise with the maximum risk of secondary salinization of irrigated lands, viscoelastic state with high stickiness and resistance to mechanical tillage, high soil compression and compaction by loads, high risks of landslides, water erosion. |

**Table 3.** *Cont.*

| Five Groups of Main FAO/USDA Soil Texture Classes | | | | | I-Sands, Silty Sands, II-Loamy Sands, Sandy Loams, III-Loams, Silty Loams, Sandy Clayey Loams, IV-Silty Loams, Clayey Loams, Fine Loams, Sandy Clays, V-Clays, Silty Clays, Clayey Loams, Silty Clayey Loams. |
|---|---|---|---|---|---|
| I | II | III | IV | V | |
| Physical Quality Index $W/W_s$ | | | | | Environmental Comments: |
| 0.15–0.25 | 0.30–0.45 | 0.35–0.55 | 0.50–0.70 | 0.60–0.80 | Available for plants water and low non-productive losses (infiltration, evaporation), moderate biological activity (basal respiration, intensity of biodegradation of organic matter, gas emission), viscoelastic state, or fragile-elastic state near the lower limit of the range, optimal for mechanical tillage and formation of granular soil aggregates, maximum concentration of water-soluble substances in soil solution with optimal mineral nutrition of roots, excluding saline soils. |
| 0.10–0.15 | 0.15–0.30 | 0.20–0.35 | 0.30–0.50 | 0.40–0.60 | Poorly available water for plants, a sharp decrease in turgor and productivity, low biological activity (intensity of biodegradation of organic matter, gas emission), very low hydraulic conductivity with disappearance of hydrostatic pressure transmission and transport of water-soluble substances, elastic-fragile state with high soil bulk density and resistance to mechanical tillage. |
| <0.10 | <0.15 | <0.20 | <0.30 | <0.40 | Non-available water for plants in the soil, plant death during prolonged drought, near-zero basal respiration and gas emissions, microbial conservation in an inactive form, non-susceptible to mechanical tillage fragile state with very intensive dusting of the soil surface, aeolian mobility for coarse-textured soils, the threat of dust and sand storms, partial or complete hydrophobization of the surface during the drought period with low initial absorption of rain water. |

Generally, the thermodynamic approach of Voronin [7] and its method of empirical secants for WRC allows quite adequately (within confidence intervals of variation) to estimate the main characteristic points (FC, MMW, WP, MAW) of WRC as the soil physical quality diagrams for most texture classes of Eurasian soils (Table 2). An exception is coarse-textured soils (groups I, II), where it is better to use the method [8] by capillary rise data for estimating FW, and for WP-correction by Shaw [50]. The main future challenges are, in our opinion, in the development of fundamental WRC models that combine the surface (disjoining pressure) and capillary mechanisms of water retention and take into account the dependence of interfacial interactions not only from the solid-base matrix, but also from the composition and concentration of the liquid phase. An equally important task is to take into account the capillarity rupture effect and the finite nature of the capillary rise of soil water and dissolved substances in computer models of the energy-mass exchange of soils within the environment. Solving both problems will improve the quantitative assessment of water retention, plant root nutrition, and the risk of salinization in the soils of arid regions, where the physical quality remains the least suitable.

## 4. Conclusions

The physical quality of the soil is determined by the interaction of its liquid, solid, and gas phases. Along with the traditionally studied capillarity, surface forces, and mechanisms, in particular, Deryagin's disjoined pressure mechanism makes a significant contribution to this interfacial interaction. The surface energy mechanism of the disjoined pressure provides aggregate stability of colloid-dispersed soil complex particles and water absorption (swelling) in the range of sorption and film moisture in the form of a linear (in pF

coordinates) WRC section, varying from 0.1 $W_s$ in sands to 0.6–0.85 $W_s$ in loams and clays. This mechanism is largely controlled by dynamic factors of the Debye thickness of the double electric layer (the charge of exchange ions and their concentration in the soil solution), which can be the cause of negative changes in the physical quality of finely dispersed soils during relatively small impacts on their liquid phase (salinization, soluble chemicals application). The improved thermodynamic approach uses soil physical quality diagrams combining water retention curves and fixed water pressure (potential) lines to determine critical crossover points in WRC. These points (maximum capillary-sorption water capacity, capillary rupture point or maximum molecular water capacity, wilting point, maximum adsorption water capacity) mark changes in physical forces and mechanisms that control water retention, dispersity, aggregate stability of the soil structure, and technological properties of the soil physical system depending on its water content. A simplified assessment of the physical quality of the main texture classes of Eurasian soils of different genesis can be carried out on the basis of a scaling table, where the soil water content at critical points is normalized by maximum value in the state of saturation.

**Funding:** This research was funded by The Russian Foundation of Basic Research, grant number 19-29-05006 (theoretical developments, Eurasian soil WRC database, and scientific equipment), and by The Russian Scientific Foundation, grant number 19-77-30012 (experimental analysis of the total water potential, and scientific equipment).

**Institutional Review Board Statement:** Not applicable.

**Informed Consent Statement:** Not applicable.

**Data Availability Statement:** Not applicable.

**Acknowledgments:** The author sincerely thanks Thomas Forkan (National University of Ireland Galway, PhD) for the assistance in spelling and technical preparation of the manuscript.

**Conflicts of Interest:** The author declares the absence of any competing financial interests or personal relationships that could have appeared to influence the work reported in this paper.

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
