# Peer review of "Thermodynamic Concept of Water Retention and Physical Quality of the Soil"

_agronomy, doi:10.3390/agronomy11091686_

Round 1
Reviewer 1 Report
This manuscript adheres to the journal’s standards. The research meets the applicable standards for the research integrity. The article does not adhere to appropriate reporting guidelines and community standards for data availability: the complete raw database is not yet made completely available in a public repository, such as Zenodo, for instance. The research output, in terms of novelty, scores good uniqueness. The level of clarity is above the threshold of acceptability. The paper does not fully discuss the limitations of the approach and potential biases due to the assumptions made. Potentially, its potential impact upon the international scientific community of reference is discrete. Experiments, statistics, and other analyses are performed to a sound technical standard and are described in detail. Conclusions presented are innovative. The article is presented in a intelligible manner. This work has discrete impact and does add a new approach to the knowledge base.
Mandatory points:
1>The name of a soil, especially when detailed, gives a synthetic but irreplaceable description of soil, its main properties, and perhaps more importantly, the processes that formed it. Also, the soil name enables one to make a posteriori inferences on aspects not directly taken into account in the context of a work. (https://doi.org/10.1021/acs.est.9b03050). Please, provide the soil classification according to the World Reference Base for Soil Resources, update 2015.
IUSS Working Group WRB. 2015. World Reference Base for Soil Resources 2014, update 2015 International soil classification system for naming soils and creating legends for soil maps. World Soil Resources Reports No. 106. FAO, Roma IT EU, 192 p.
2>Data sharing is the practice of making data used for scholarly research available to other investigators. Many funding agencies, institutions, and publication venues have policies regarding data sharing because transparency and openness are considered by many to be part of the scientific method. Public repositories, such as for instance Zenodo (general-purpose open-access repository operated by CERN) allows researchers to deposit mainly raw data sets, research software, and any other research related research items granting individual submissions with a persistent digital object identifier (DOI) which makes the stored items easily citeable. The practice of filing the original database, highly encouraged in recent years, is essential for any scientific work published after 2020.
Author Response
Dear Reviewer!
Thank you very much for your positive assessment of the manuscript and valuable comments on improving it. The main comment concerns the use of the WRB soil classification. I have changed the names that do not correspond to the latest version of the WRB classification and provided a link to this classification. However, one must understand that this is only an approximate translation. Unfortunately, objective difficulties do not allow us to accurately translate the Russian names of soils into the WRB classification, since these are fundamentally different classification systems. WRB is a descriptive classification, the Russian school operates with a genetic classification. I will allow myself an analogy. Tomatoes can be yellow, red and green in color (WRB classification). The tomato can be unripe, ripe and rotten by its state (Russian classification). How do you compare them? It is most likely that an unripe tomato is green, and a ripe tomato is yellow or red. Rotten can be of any color. So it is with soil transfer, especially when there are no (lost) exact boundaries of horizons and field morphological descriptions. Our sod-podzolic soils can be Luvisols, Retisols, and if sandy, then Arenosols. Our brown and gray forest soils can be Phaeozems, Umbrisols. It is very difficult to choose the name of our «Solod» or our «Zheltozem» (Caucasian yellow soil of tea plantations). In general, I tried to do this, but, I repeat, it is impossible to achieve an exact translation.
On the second remark. I am not ready to provide now the full version of the WRC-database in English, since this is a separate large work. In addition, I'm not sure if it should be publicly available, since such data also have commercial value, in particular, I plan to apply for a separate research grant on this topic. I'm not sure, is it possible to put now in Zenodo the English fragment of this database prepared by me (I will try after answering both reviewers). If not, then I can send it to you, as a reviewer, for private review and evaluation, along with the revised manuscript.
Thanks again for your feedback. Sincerely yours, Andrey Smagin

Reviewer 2 Report
Agronomy-1312919-peer-review-v1:
This article studies, describes, and interprets the physical quality of soil on a thermodynamic basis. Instead of the soil quality index introduced by Dexter on a practical basis, the author intends to develop a soil quality characteristic supported by thermodynamic theory. The evolution of soil moisture content and water potential over the entire moisture range was measured by high-speed centrifugation, and the thermodynamic potential was determined by temperature of the water-ice phase. According to the ‘Russian school’, it analyzes and describes the attractive and repulsive forces in detail between the solid skeleton, swellable clay, organic matter and water in a soil polydisperse system. The author interprets the molecular details of soil water retention as a function of water content as a balance of attractive and repulsive forces on an energetic basis. Synthesizes is between the theories on the water holding capacity of Russian and Western soils. By introducing critical water content, it gives a physical interpretation of Dexter’s WRC inflection point slope. I accept the theoretical findings, structure and English language correctness of the article. However, I miss the comment of the papers published on the same or closely similar concepts (see the Major comment).
Major comment:
Given that a method for the water holding capacity of soils with thermodynamic concepts, which has also appeared in the international literature, should also be evaluated in this paper (e.g. Braudeau, E., Assi, A. T., Boukcim, H., and Mohtar, R. H. (2014). Physics of the soil medium organization part1: thermodynamic formulation of the pedostructure water retention and shrinkage curves. Front. Environ. Sci. 2:4. doi: 10.3389/fenvs.2014.00004).
Minor comments:
62: line 62 is missing;
162-166: The used soil classification system is not given.
255: The Hamaker-Lifshitz equation reference should be given.
316: A reference or an explanation of BET estimates is needed.
304: The Russian letter i shall be replaced by English.
305: Both Equations (8) and (9) lack an explanation for the Sρ value.
372: The Sc value estimate by eg.(8) is not clear.
378: Specifying a conversion of 1 mm = 10-3 m is unnecessary.
455: The meaning of ‘ore’ in the bracket between MSCW and FW is not known!
465: The reference of the author’s database should be given e.g. as internet link.
473: Table 1, cited on page 14, is on page 21. This needs to be corrected.
477: The value here of Dexter’s soil physical quality of group I is not that in Table 1.
481-488: The referenced Z index values ​​are missing. In a scientific article this is not allowed.
489-500: The results in this paragraph are not presented in this paper, and the cited citations are only partially available to the readers.
503: Unfortunately, Table 2 is almost at the end of the article on page 23, far from the reference point. It is advisable to change this.
541: Referring to Figure 2, a sub-diagram A, B, or C of the figure is missing.
570: As the statement "classical material" according to the cited Russian-language literature is not clear, further explanation is needed..
559-562: The captions in Figures C and D of Figure 6 are missing.
580-580: In Table 3 in the 5th category: “Non-available water for plants in the soils, ---
and aeolian mobility for coarse textured soils” ….
With the comments listed above I suggest a major revision of the manuscript.
Author Response
Dear Reviewer!
I sincerely thank you for your professional and positive feedback on my manuscript and your valuable comments on correcting it. I agree with all the comments and below I give the answers to the points of the review.
According to the main comment. Thank you so much for it.. I was friends with Peter Berezin, a professor of our department and, like me, a pupil and collaborator of professor Anatoly Voronin. Unfortunately, both of them have already passed away. Peter Berezin in Russia was first, who began to develop the approach that you are writing about.. I have given in the text (lines 87-94) and in the reference list corresponding references to these works, as well as to the excellent work doi: 10.3389 / fenvs.2014.00004, which was unknown to me before your feedback. All these works are extremely interesting for the relationship between soil water retention and structure dynamics. We follow parallel paths here. I am trying to consider the dynamics of dispersion based on the classical DLVO-theory of colloids stability (doi: 10.1134 / S1064229318070098). Petrer Berezin and professor Braudeau. et al (doi: 10.3389 / fenvs.2014.00004) do this through the dynamics of specific pore volume of pore space. I also had this experience (doi: 10.1134 / S1064229320070157), but I do not use it in this manuscript, limiting myself to a simpler “REV approach” in the terminology of Braudeau, et al. It seems to me that in the future it is necessary to combine both approaches and develop a detailed applied scale of the physical quality of soils, including swelling soils with variable pore space, the processes of compaction and physical fusion, which professor Berezin began to do in Russia (Berezin, P.N. Diagnostics of potential and actual compaction following physical criteria. Pochvovedeniye 5, 65–75. (1990).
Minor comments:
Line 62 deleted
A link to the WRB-classification is given and the names of soils are corrected
The Hamaker-Lifshitz equation is contained in Russian textbooks of colloidal chemistry. I have given a link to sources from the manuscript referencing sheet where this equation is discussed (line 203).
Decoding and reference to the BET estimate of the specific surface area is given on lines 255-256
The letter "and" is replaced by the union "and" in the sense
Sρ value is the product of specific surface area (S) and water density (ρ). Formulas (7-9) were corrected
The same idea about the Sc-value
1 mm = 10-3 m was removed
The meaning of ‘ore’ is a misspell. Replaced with "or"
I cannot yet publish the entire database in the open access, as I hope to receive a separate scientific grant for this work. Here is a link for reference https://yadi.sk/i/w-jxkQiNBWMVJQ
I put tables 1 and 2 in place (this is a problem with the automatic service for the Agronomy MDPI manuscript)
Z index values ​​are corrected (lines 371-386) and placed in table 1.
"489-500: The results in this paragraph are not presented in this paper". This is not quite right as the results (WRC straight segments) are shown in Figure 4-A. Line 390 now has a link to Figure 4-A.
Figure 2 with signatures containing positions A, B, C is shown on lines 229-231. The text references to Figure 2 are on lines 186,191, 196, 429.
The “classic” in science is interpreted as: “A well known and reliable procedure (material), such as a demonstration of a well-established scientific principle, may be described as classic. (https://en.wikipedia.org/wiki/Classic) This is a generally accepted term. The work [40] for Russian agronomy is classic, as, for example, the work of Sposito G. (1981) for Thermodynamics of soil water.
Lines 448-451 contain the caption to figure 6 with explanations
In Table 3 in the 5th category: “Non-available for plants water in the soils has been replaced by
“Non-available water for plants in the soils
Once again, thank you very much for your work in reviewing the manuscript.
Sincerely yours, prof. Andrey Smagin.

Round 2
Reviewer 2 Report
The revised manuscript ID: agronomy-1312919 and the author's reply I've accepted. So, I beleive that the carefully revised manuscript is now suitable for publication in Agronomy. I have one correction proposal in line 473: ...(8) (Fig. 4 - A).